# Kidney protection strategy lowers the risk of contrast-associated acute kidney injury

Chien-Boon Jong [1,2,3]*, Jui-Cheng Kuo[4], I-Chuan Lin[5]

1 Department of Internal Medicine, National Taiwan University Hospital, Hsin-Chu Branch, Hsin-Chu, Taiwan, 2 Department of Internal Medicine, National Taiwan University Hospital, Taipei, Taiwan, 3 College of Medicine, National Taiwan University, Taipei, Taiwan, 4 Department of Radiology, National Taiwan University Hospital, Hsin-Chu Branch, Hsin-Chu, Taiwan, 5 Department of Nursing, National Taiwan University Hospital, Hsin-Chu Branch, Hsin-Chu, Taiwan

* jgboon0407@gmail.com

**Data Availability Statement:** Data cannot be shared publicly because of ethical restrictions. The restrictions apply to the availability of these data, which were used under license for the current study, and so are not publicly available. Data are

## Abstract

We developed a comprehensive kidney protection strategy (KPS), which comprises left ventricular end-diastolic pressure-guided saline hydration, ultralow contrast coronary angiography, and a staged coronary revascularization procedure under suitable conditions. This study aimed to investigate KPS's effect on the risk of developing contrast-associated acute kidney injury (CA-AKI) among persons with moderate-to-advanced kidney insufficiency (KI). Seventy patients who had undergone cardiac catheterization with an estimated glomerular filtration rate (eGFR) of 15–45 mL/min/1.73 m$^2$ were investigated retrospectively. Among these, 19 patients who had received KPS and 51 who had undergone cardiac catheterization with usual care (UC) were enrolled. CA-AKI was defined as a 0.3-mg/dL increase in serum creatinine levels or dialysis initiation within 72 h after contrast exposure. The inverse probability of treatment weighting (IPTW)-adjusted cohort was analyzed according to the Mehran 2 risk categories. Patients' mean age was 73.3 ± 9.6 years; mean eGFR was 29.8 ± 8.5 mL/min/1.73 m$^2$; and median of Mehran 2 risk score, 8. Most patients presented with acute myocardial infarction (AMI) or heart failure, and one-fifth of the administered cardiac catheterizations were emergency procedures. After the IPTW adjustment, the KPS group showed a significantly lower CA-AKI risk than the UC group (4% vs. 20.4%; odds ratio 0.19, 95% confidence interval 0.05–0.66). This effect was consistent across various subgroups according to different variables, including old age, AMI, advanced KI, high-risk category, left ventricular systolic dysfunction, and multivessel disease. Conclusively, KPS may reduce the CA-AKI risk in high-risk patients with moderate-to-advanced KI who have undergone cardiac catheterization.

## Introduction

Contrast-associated acute kidney injury (CA-AKI) is linked with an increased risk of mortality and major cardiovascular events [1,2]. Over the decades, the incidence of CA-AKI after cardiac catheterization and its associated mortality has not declined [3]. The incidence of CA-AKI

available from the National Taiwan University Hospital Ethics Committee (contact via ntuhrec@ntuh.gov.tw) for researchers who meet the criteria for access to confidential data.

**Funding:** This study was supported by grants from the National Taiwan University Hospital, Hsinchu branch (112-HCH106, 113-HCH092) (https://www.hch.gov.tw/) and National Taiwan University Hospital (113-GH041)(https://www.ntuh.gov.tw/). CBJ received these Grant awards. The funders had no role in the study design, data collection and analysis, publication decision, or manuscript preparation.

**Competing interests:** The authors have declared that no competing interests exist.

increases with the severity of kidney insufficiency (KI) and higher risk scores as per the prediction model [4,5].

Notably, several measures have been proposed to prevent CA-AKI during coronary angiography (CAG); the most desirable is the periprocedural administration of intravenous isotonic saline [6–8]. Left ventricular end-diastolic pressure (LVEDP)-guided isotonic saline hydration has recently been suggested as an alternative to saline hydration with a constant infusion rate. The efficacy of this method has been shown in patients with KI and those at a high risk of CA-AKI [6]. In addition, administering an ultralow contrast amount, less than the estimated glomerular filtration rate (eGFR) value, was proposed to minimize contrast administration during CAG in patients with KI [9]. This ultralow contrast CAG (ULCAG) strategy correlates with a lower CA-AKI risk [10]. However, reports on these methods' synergistic effects are rare.

The causal association of contrast agents with AKI remained controversial in recent years. The role of contrast agents in the development of AKI may be overestimated, especially in cases of multi-comorbidity risks [4,11]. Therefore, this study stratified the risk of AKI according to a novel predictive model that included eight clinically important variables, with the exception of the contrast volume [4]. Herein, a comprehensive kidney protection strategy (KPS) that combines the concepts of LVEDP-guided isotonic saline hydration, ULCAG, and a staged coronary revascularization procedure under suitable conditions was introduced. This study aimed to investigate the effect of KPS on CA-AKI risk in a cohort with moderate-to-advanced KI and high-risk conditions.

## Materials and methods

### Study population and ethical approval

In our institute, comprehensive KPS has been conducted preceding zero contrast percutaneous coronary intervention (PCI) since November 2019 [12]. All patients referred to our study team for CAG underwent KPS, while comorbidity with an eGFR of <45 mL/min/1.73 m$^2$. In contrast, patients with KI who were referred to others interventionist and underwent similar procedures, but did not receive KPS during CAG were categorized into the usual care (UC) group. This study enrolled patients from two databases: those registered in the catheterization laboratory database and those who underwent KPS during CAG performed by study team. The enrollment periods of the two databases overlapped, and 17 patients from the catheterization laboratory database were reclassified into the KPS group because they had undergone KPS during CAG (S1 Fig). At our institute, iodixanol, a nonionic, iso-osmolar contrast medium, was administered to patients with KI during CAG. The inclusion criteria were iodixanol administration, an eGFR of <45 mL/min/1.73 m$^2$ during CAG, and an indication for CAG due to acute coronary syndrome, chronic coronary syndrome, or heart failure. Patients with contrast exposure within 1 week before the index procedure and those with cardiac arrest or who received mechanical circulatory support before or during CAG were excluded from the analysis. Because nearly half of the patients with an eGFR of <15 mL/min per 1.73 m$^2$ had initiated hemodialysis before CAG with UC, those with an eGFR of <15 mL/min per 1.73 m$^2$ were also excluded from the final analysis. The other exclusion criteria are listed in S1 Fig. Multiple contrast administrations in separate coronary procedures were counted only at the first procedure. Ultimately, 51 patients who received UC and 19 who received KPS were enrolled in this study. All data were retrospectively collected from electronic medical records and our catheterization laboratory database. These data were assessed from 2$^{nd}$ September 2023 to 30$^{th}$ October 2023. The initial data were identifiable to personal information during collection, and which were anonymized after collection. The formula for the modification of diet in renal disease was used to calculate eGFR [13]. This study was approved by the

institutional review board (202307197RINA) of the National Taiwan University Hospital, and the need for informed consent was waived. This study was performed in accordance with the latest version of the Declaration of Helsinki (2013) and all relevant regulations.

## Protocol of comprehensive KPS

Notably, all patients who underwent KPS received 1.5 mL/kg/h isotonic saline hydration 1 h before the scheduled CAG or upon arrival at the catheterization room in cases of emergency procedures. LVEDP was measured at the initiation of the procedure, and the saline infusion rate was titrated based on it. In brief, 5 mL/kg/h of intravenous saline infusion was used for an LVEDP of <13 mmHg, 3 mL/kg/h for an LVEDP of 13–18 mmHg, and 1.5 mL/kg/h for an LVEDP of >18 mmHg [6]. This infusion rate commenced before contrast exposure, during CAG, and continued for another 4 h after CAG or ad hoc PCI [6]. A targeted maximum contrast volume equal to or less than the latest numeric eGFR value was used to minimize contrast administration during CAG [9]. At each CAG shot, 2 mL of contrast media diluted in 1–2 mL saline was administered using biplane cine imaging. Two shots of the left coronary artery and one of the right coronary artery can be used to evaluate the extent of all epicardial coronary artery stenoses in most patients. Fractional flow reserve assessment was routinely used in case of intermediate stenosis or ambiguous lesions, either during the index or staged procedure [14,15]. If clinically suitable, zero contrast PCI, coronary bypass surgery, or valve replacement surgery was performed 7 days after contrast exposure [12]. Furthermore, intravascular ultrasonography was used routinely during PCI, particularly in primary PCI [16].

## UC strategy

In contrast, patients receiving UC were hydrated with an isotonic saline infusion at a constant rate of 0.5 or 1 mL/kg/h in those with or without heart failure (HF) before and after 12 h of contrast exposure [8]. Notably, the strategy for CA-AKI prevention in UC group was not standardized in all patients, which reflects the real-world situation in the daily catheterization room. Some patients may not have received saline hydration because of concerns regarding fluid overload. Minimum contrast volume administration and decision on the timing of coronary revascularization were not standardized; they depended on the operator's discretion.

## Definition of CA-AKI

Serum creatinine follow-up on days 1 and 3 after contrast exposure was performed in most patients receiving KPS. The highest post-CAG creatinine level was recorded to determine the CA-AKI incidence. The same methodology was applied to patients receiving UC. Patients without serum creatinine follow-up or out-of-follow-up time intervals were excluded from the analysis (S1 Fig). CA-AKI was defined as a >0.3-mg/dL increase in serum creatinine concentration or hemodialysis initiation within 72 h after contrast exposure [4,17]. The baseline serum creatinine value was measured on days 0–3 before the index procedure.

## Mehran 2 risk-predicting model for CA-AKI

The Mehran 2 risk-predicting model for CA-AKI, introduced in 2021 by Mehran et al., is the latest CA-AKI risk model in the contemporary PCI era [4]. This study applied Model 1 of the proposed risk models, which included eight clinical variables readily available before CAG. These variables were the clinical presentation of acute coronary syndrome, lower eGFR, left ventricular ejection fraction (LVEF) <40%, diabetes, hemoglobin <11 g/dL, basal glucose ≥150 mg/dL, congestive HF on presentation, and age >75 years. A weighted risk score was

proposed for each variable, and the summed risk score was categorized into four groups based on the threshold of the corresponding risk score. The corresponding thresholds of the risk score for each risk category were ≤2, 3–7, 8–11, and ≥12 in the low-, moderate-, high-, and very high-risk categories, respectively. The risk of CA-AKI increases across these four risk categories [4].

### Statistical analysis

The baseline characteristics of patients in the KPS and UC groups were compared using an independent sample t-test for continuous variables, the Mann–Whitney U-test for skewed continuous variables (contrast, the ratio of contrast volume to eGFR value, and Mehran 2 risk score), and Fisher's exact test for categorical variables. The CA-AKI risks predicted by Mehran et al. and this study's actual CA-AKI risks in different Mehran 2 risk categories were compared using the independent proportion z-test [4]. The association between baseline characteristics and CA-AKI risk was evaluated using the Firth logistic regression model due to the limited sample size (n = 70) and rare occurrence (n = 11) [18]. A series of univariate Firth logistic regression analyses were conducted, and variables with a p-value <0.15 were further included in the multivariate model [19]. Since substantial differences existed between the Mehran 2 risk score and the risk category between the KPS and UC groups, an inverse probability of treatment weighting (IPTW)-adjusted cohort with average treatment effect based on propensity score was created. The risk of CA-AKI between groups was further compared in the IPTW-adjusted cohort. The analysis was stratified using several prespecified subgroup variables, including age (<75 vs. ≥75 years), clinical presentation (acute myocardial infarction [AMI] vs. HF), eGFR at cath (<30 vs 30–45), Mehran 2 risk category (mild/moderate vs. high/very high), LVEF (<40% vs. ≥40%), and multivessel disease. Statistical significance was set at a two-sided P<0.05. All analyses were performed using SAS version 9.4 (SAS Institute, Cary, NC, USA).

### Results

The mean age was 73.3 years (standard deviation [SD] = 9.6 years), and the mean eGFR was 29.8 ± 8.5 mL/min/1.73 m$^2$. Approximately 50% of the patients had eGFR<30 mL/min/1.73 m$^2$, and 70% had diabetes mellitus. Furthermore, 50% of the patients had a clinical presentation of AMI, and another 44% had HF. A third of the patients had an LVEF of <40%, and 74% were diagnosed with multivessel coronary artery disease. A fifth of the CAGs performed were emergency procedures, and a quarter of the patients received ad hoc PCI. The KPS and UC groups had comparable baseline clinical conditions, cardiac performance, and catheterization profiles. However, there were more multivessel coronary artery disease cases in the UC group (Table 1).

Along with the protocol of KPS, the KPS group had a lower contrast volume exposure; all had a contrast volume/eGFR ratio of ≤1 [9]. The number of patients who received isotonic saline hydration was twice as high in the KPS group as in the UC group (89.5% in KPS vs. 43.1% in UC; P<0.001). However, the rate of staged coronary revascularization was not significantly different between the two groups (Tables 1 and 2).

Data are presented as frequency (percentage), mean ± standard deviation, or median [25th percentile, 75th percentile].

Overall, the median Mehran 2 risk score was 8 (interquartile range [IQR]: 6–11.25), and approximately 51% of the patients were classified into high- and very high-risk categories. Notably, the incidence of CA-AKI was increased with the risk category, and the risk across different categories was consistent with the initial report by Mehran et al. (S2 Fig) [4].

**Table 1. Baseline demographics and clinical characteristics of the patients.**

| Variable | KPS (n = 19) | UC (n = 51) | P value |
|---|---|---|---|
| Age, years | 73.4 ± 10.6 | 73.3 ± 9.3 | 0.977 |
| Female sex | 5 (26.3) | 19 (37.3) | 0.572 |
| Body mass index, kg/m$^2$ | 23.8 ± 2.7 | 24.6 ± 3.3 | 0.380 |
| Smoke | 9 (47.4) | 23 (45.1) | 1.000 |
| Diabetes mellitus | 14 (73.7) | 35 (68.6) | 0.776 |
| Hypertension | 14 (73.7) | 41 (80.4) | 0.531 |
| Statin at discharge | 15 (78.9) | 41 (80.4) | 1.000 |
| ARB at discharge | 10 (52.6) | 33 (64.7) | 0.414 |
| Clinical presentation | | | 0.900 |
| STEMI | 2 (10.5) | 7 (13.7) | |
| NSTEMI | 6 (31.6) | 19 (37.3) | |
| Heart failure | 10 (52.6) | 21 (41.2) | |
| US/CCS | 1 (5.3) | 4 (7.8) | |
| Laboratory data | | | |
| Hemoglobin, g/dL | 10.9 ± 2.3 | 11.3 ± 2.3 | 0.526 |
| Serum creatine at cath, mg/dL | 2.4 ± 0.7 | 2.3 ± 0.7 | 0.696 |
| Mean eGFR at cath, mL/min/1.73 m$^2$ | 29.1 ± 8.1 | 30.0 ± 8.7 | 0.707 |
| eGFR 30–45 at cath, mL/min/1.73 m$^2$ | 10 (52.6) | 27 (52.9) | 1.000 |
| eGFR < 30 at cath, mL/min/1.73 m$^2$ | 9 (47.4) | 24 (47.1) | |
| Baseline eGFR[a], mL/min/1.73 m$^2$ (n = 35) | 36.9 ± 11.7 (n = 5) | 36.9 ± 11.7 (n = 30) | 0.680 |
| LDL, mg/dL (n = 65) | 73.3 ± 35.8 | 92.8 ± 42.5 | 0.091 |
| HbA1c, % (n = 61) | 7.4 ± 1.8 | 6.8 ± 1.4 | 0.140 |
| Cardiac performance | | | |
| Congestive heart failure | 16 (84.2) | 33 (64.7) | 0.148 |
| LVEF <40% | 9 (47.4) | 15 (29.4) | 0.171 |
| Valvular heart disease | 5 (26.3) | 10 (19.6) | 0.531 |
| Cardiac catheterization | | | |
| Femoral access | 5 (26.3) | 20 (40.0) | 0.403 |
| Emergency procedure | 4 (21.1) | 11 (21.6) | 1.000 |
| Ad hoc PCI | 2 (10.5) | 15 (29.4) | 0.127 |
| Multivessel disease | 10 (52.6) | 42 (82.4) | 0.028 |

Abbreviations: KPS, kidney protection strategy; UC, usual care; eGFR, estimated glomerular filtration rate; ARB, angiotensin II receptor antagonist; STEMI, ST elevation myocardial infarction; NSTEMI, non-ST elevation myocardial infarction; US, unstable angina; CCS, chronic coronary syndrome; LDL, low-density lipoprotein; HbA1c, glycated hemoglobin; LVEF, left ventricular ejection fraction; PCI, percutaneous coronary intervention

Data are presented as frequency (percentage) or mean ± standard deviation.

[a]Determined by eGFR 3–6 months prior to coronary angiography.

The initial Mehran risk category and risk score (pre-IPTW) are numerically higher in the KPS group than in the UC group (left panels in Fig 1A and 1B). However, the initial CA-AKI risk was numerically lower in the KPS group than in the UC group (5.3% in the KPS group vs. 19.6% in the UC group; P = 0.267) (Tables 2 and (S1). After adjusting for the Mehran risk category with IPTW, the Mehran risk category and risk score were well balanced (right panel in Fig 1A and 1B). The CA-AKI risk was significantly lower in the weighted KPS group than in the weighted UC group (CA-AKI risk after IPTW adjustment: 4.0% in the KPS group vs. 20.4% in the UC group; odds ratio 0.19, 95% confidence interval 0.05–0.66; P = 0.009) (Fig 2).

**Table 2. Characteristics of KPS and CA-AKI in the patients.**

| Variable | KPS (n = 19) | UC (n = 51) | P value |
|---|---|---|---|
| Contrast, mL | 7.5 [5.0, 11.0] | 25.0 [20.0, 65.0] | <0.001 |
| CG ratio | 0.24 [0.20, 0.44] | 0.96 [0.60, 1.81] | 0.003 |
| CG ratio ≤1 | 19 (100.0) | 27 (52.9) | <0.001 |
| Acetylcysteine | 17 (89.5) | 11 (21.6) | <0.001 |
| Hydration | 17 (89.5) | 22 (43.1) | 0.001 |
| Hydration volume, mL (n = 39) | 694.7 ± 248.6 | 670.5 ± 365.0 | 0.816 |
| CA-AKI | 1 (5.3) | 10 (19.6) | 0.267 |
| Kidney function after day 1–3 of contrast exposure | | | 0.380 |
| Improved | 6 (31.6) | 16 (31.4) | |
| Stable | 12 (63.2) | 25 (49.0) | |
| AKI (deteriorated) | 1 (5.3) | 10 (19.6) | |

Abbreviations: KPS, kidney protection strategy; UC, usual care; CG ratio, contrast/estimated glomerular filtration rate ratio; CA-AKI, contrast-associated acute kidney injury.

Notably, all subgroups with nonsignificant interactions in the two groups showed consistent results in terms of CA-AKI risk (Fig 2).

## Discussion

This study demonstrated that patients receiving comprehensive KPS had higher saline hydration and lower contrast exposure during cardiac catheterization. Combining these protective effects may lower CA-AKI risks in patients with KI, particularly those with advanced KI. To the best of our knowledge, this is the first study to show the cumulative effect of LVEDP-guided hydration and ULCAG in reducing CA-AKI in patients with concomitant high-risk conditions and KI.

The Mehran 2 risk-prediction model for CA-AKI was recently introduced and derived from a large cohort in the contemporary PCI era. Mehran initially proposed two risk

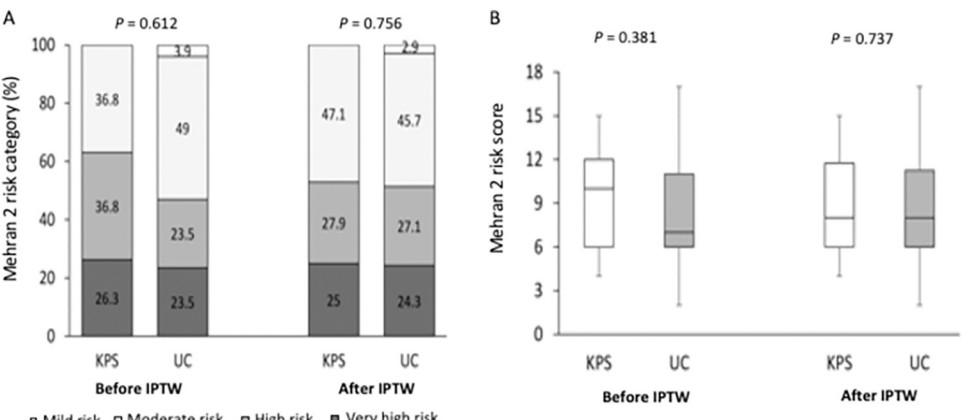

**Fig 1.** Mehran 2 risk category (A) and risk score (B) in patients in the KPS and UC groups before and after IPTW adjustment. The initial Mehran risk category and risk score (pre-IPTW) are numerically higher in the KPS group than in the UC group (left panels in Figures A and B). After adjusting for the Mehran risk category with IPTW, the Mehran risk category and risk score are well-balanced (right panels in A and B). KPS, kidney protection strategy; UC, usual care; IPTW, inverse probability of treatment weighting.

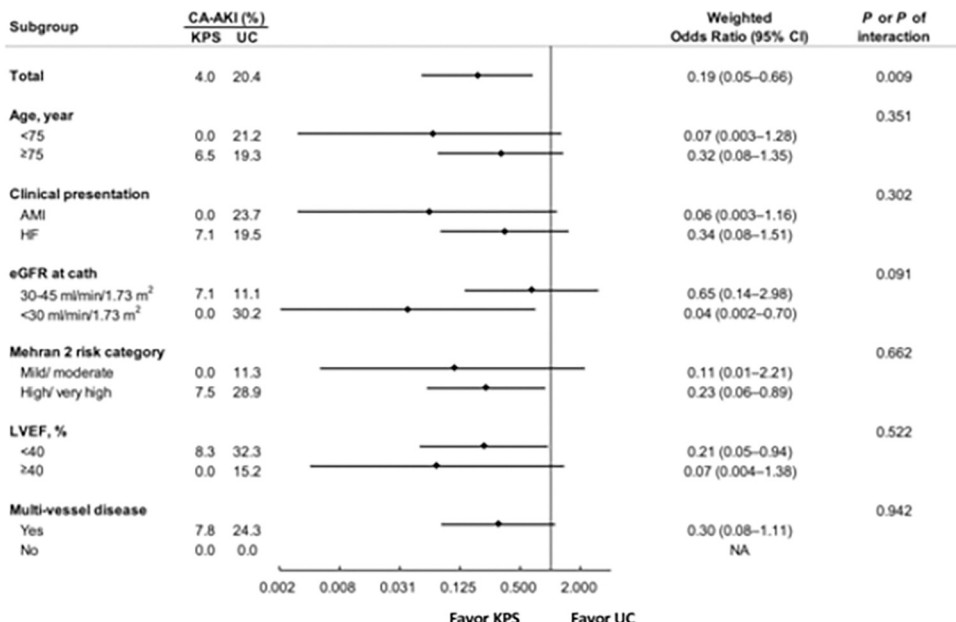

**Fig 2. CA-AKI risk in patients receiving KPS versus UC stratified using subgroup variables in the IPTW-adjusted cohort.** CA-AKI risk is significantly lower in the weighted KPS group than in the weighted UC group. There are consistent results regarding CA-AKI risk between the two groups across all subgroups, with nonsignificant interactions. KPS, kidney protection strategy; UC, usual care; CA-AKI, contrast-associated acute kidney injury; IPTW, inverse probability of treatment weighting; AMI, acute myocardial infarction; HF, heart failure; CKD, chronic kidney disease; LVEF, left ventricular ejection fraction; NA, not applicable.

prediction models, with Model 2 having a slightly higher discriminative power than Model 1 [4]. Model 1 included eight common clinical variables readily available before the coronary procedure, while Model 2 had four additional procedural variables acquired after the procedure. Model 1 was used for risk prediction in this study due to the missing procedural variables. The most impactful variables on CA-AKI risk were AMI and advanced KI in the Mehran 2 risk model. Notably, both variables were common comorbidities in our cohort study, accounting for approximately 49% each. Half of the study population had a high to very high risk of CA-AKI, and the actual risk of CA-AKI was consistent with that shown in Mehran's study group's initial report across various risk categories [4]. This result revealed the generalizability of the Mehran 2 risk model in the Chinese population and vice versa, representing an unintentional population selection in our study cohort.

The concept of the KPS is to manage multiple modifiable risks, simultaneously reducing the risk of CA-AKI; this has seldom been reported previously [6,7,9,17]. Moreover, maximizing volume expansion, minimizing contrast exposure, and introducing zero-contrast PCI are fundamental to our KPS [7,9,12]. The efficacy and safety of LVEDP-guided saline hydration have been reported. However, the initial reporting protocol excluded patients with acute HF, severe valvular disease, and emergency procedures [6]. Our study team modified the initial protocol with hydration at a constant rate of 1.5 mL/kg/h 1 h before the procedure or upon arrival at the catheterization room during emergency procedures [20]. This modification was based on the fact that most of the patients in the KPS group had concomitant HF or high estimated LVEDP from preprocedural echocardiography; therefore, the need for information on estimated LVEDP before CAG in emergency conditions was largely nonexistent. This modified protocol was feasible for emergency procedures and in patients with medically controlled severe valvular heart disease. Furthermore, invasive LVEDP measurement is a common

procedure in the daily catheterization room, and the long-standing experience in LVEDP-guided fluid management in patients with HF made the LVEDP-guided saline hydration protocol more comfortable among cardiologists, particularly under HF conditions [21,22]. In contrast, the rate of hydration was significantly lower in the UC group, wherein two-thirds of patients had a history of HF. Most of the non-hydrated patients may be concerned about fluid overload, which reflects the real-world situation in daily catheterization room. The lower rate of hydration in the UC group may have led to higher CA-AKI in this high-risk population. This study showed that KPS reduced the risk of CA-AKI by approximately 80% compared with the UC strategy, and this result was consistent with a previous report suggesting that LVEDP-guided saline hydration reduced CA-AKI by 68% compared with UC [6]. The additional effect of KPS may have been partly caused by the ULCAG strategy, which minimized contrast administration during the coronary procedure.

For over a decade, it has been reported that an increased contrast volume-to-eGFR ratio elevated the risk of CA-AKI while deteriorating kidney function [5]. However, ULCAG was recently recommended in a cohort of patients with advanced KI who planned to receive zero contrast PCI [9]. Moreover, the correlation between ULCAG and lower CA-AKI risk has been reported in patients with advanced KI [10]. Currently, all patients in the KPS group received ULCAG, and one-third of patients underwent staging procedures, such as zero contrast PCI and cardiac surgery, a few days later[12]. Staging the procedure can avoid large amounts of contrast exposure, allowing the contrast media a 1-week "washout" period [7]. These comprehensive strategies may provide additional benefits in the KPS group to reduce CA-AKI risk in high-risk populations since only half of the patients met the definition of ULCAG in the UC group. Minimizing contrast exposure with ULCAG and staging procedures is reasonable; however, this strategy has seldom been used in recent clinical trials for CA-AKI prevention [6,17].

This study has few limitations. First, the influence of residual confounding factors and selection and indication biases are unavoidable in observational studies. In addition, this study's small sample size limited further confounder adjustment, and the analysis was conducted on the IPTW-adjusted cohort using only the Mehran 2 risk category. However, in the initial cohort, the KPS and UC groups had comparable baseline clinical conditions, laboratory profiles, cardiac performance, and catheterization profiles. Third, half of the patients did not have baseline eGFR data, which implies that some of the patients may have developed AKI during CAG. However, the follow up kidney function after CAG showed that nearly one-third of the patients had improved kidney function in both groups without a significant difference. Based on this result, we assumed that the proportion of AKI in both groups might be similar and have less impact on the risk of CA-AKI. Fourth, fluctuations in post-CAG serum creatinine levels within the 72-hour monitoring period may occur, and some data might go undetected, particularly in situations without frequent monitoring. This may lead to an underestimation of the incidence of CA-AKI. Finally, one-third of the population receiving UC was excluded from the final analysis due to unavailable data on serum creatinine value during the follow-up period of 3 days. Nevertheless, 21.4% of the excluded population met the criteria for CA-AKI, with serum creatinine values checked after 72 h of contrast exposure. Furthermore, the incidence rate of CA-AKI in the UC group was consistent with Mehran et al.'s report and other clinical trials [4,6,17]. Therefore, it is assumed that these exclusion criteria did not confound this study's results.

## Conclusion

Comprehensive KPS may reduce the risk of CA-AKI in high-risk patients with moderate-to-advanced KI who undergo coronary catheterization. Due to this study's single-center setting and small sample size, further multicenter trials are necessary to evaluate its efficacy.

## Supporting information

**S1 Table. Associated factors of contrast-associated acute kidney injury.**
(DOCX)

**S1 Fig. Workflow of the study protocol.**
(DOCX)

**S2 Fig.** Predicted and actual CA-AKI risks in different Mehran 2 risk categories in the entire cohort (a), KPS group (b) and UC group (c).
(DOCX)

## Author Contributions

**Conceptualization:** Chien-Boon Jong.

**Data curation:** Jui-Cheng Kuo, I-Chuan Lin.

**Formal analysis:** Chien-Boon Jong.

**Funding acquisition:** Chien-Boon Jong.

**Investigation:** Chien-Boon Jong, Jui-Cheng Kuo, I-Chuan Lin.

**Methodology:** Chien-Boon Jong.

**Project administration:** Chien-Boon Jong, Jui-Cheng Kuo, I-Chuan Lin.

**Resources:** Chien-Boon Jong.

**Writing – original draft:** Chien-Boon Jong.

**Writing – review & editing:** Jui-Cheng Kuo, I-Chuan Lin.

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
