## [Decision Letter · Decision Letter 0]

1 Jul 2024

PONE-D-24-15647Kidney protection strategy lowers the risk of contrast-associated acute kidney injuryPLOS ONE

Dear Dr. Jong,

Thank you for submitting your manuscript to PLOS ONE. After careful consideration, we feel that it has merit but does not fully meet PLOS ONE’s publication criteria as it currently stands. Therefore, we invite you to submit a revised version of the manuscript that addresses the points raised during the review process.

We look forward to receiving your revised manuscript.

Kind regards,

Eyüp Serhat Çalık

Academic Editor

PLOS ONE

Journal Requirements:

3. Ethics statement only appears at the end of the manuscript:

Your ethics statement should only appear in the Methods section of your manuscript. If your ethics statement is written in any section besides the Methods, please move it to the Methods section and delete it from any other section. Please ensure that your ethics statement is included in your manuscript, as the ethics statement entered into the online submission form will not be published alongside your manuscript.

Additional Editor Comments:

I congratulate the authors for their work on this important topic. The strategy you propose seems to be an important method to prevent contrast nephropathy, but it needs to be supported by prospective randomized studies, I hope it will set an example for larger studies. Your manuscript has been reviewed by three esteemed reviewers, their recommendations are as follows. We look forward to receiving your revised manuscript, including your point-by-point responses to the suggestions and any necessary revisions. We wish you success.

Reviewers' comments:

Reviewer's Responses to Questions

**Comments to the Author**

1. Is the manuscript technically sound, and do the data support the conclusions?

Reviewer #1: Partly

Reviewer #2: Partly

Reviewer #3: Partly

2. Has the statistical analysis been performed appropriately and rigorously? 

Reviewer #1: Yes

Reviewer #2: Yes

Reviewer #3: No

3. Have the authors made all data underlying the findings in their manuscript fully available?

Reviewer #1: No

Reviewer #2: Yes

Reviewer #3: Yes

4. Is the manuscript presented in an intelligible fashion and written in standard English?

Reviewer #1: Yes

Reviewer #2: Yes

Reviewer #3: Yes

5. Review Comments to the Author

Reviewer #1: Dear editor,

first of all thank you for asking me to review this manuscript about kidney protection strategy to lower contrast induced kidney injury. To date, this is a extremely hot topic.

For the regards the content:

Can the authors be more specific about patients enrolled? Are all ST elevation myocardial infarction patients? or not?

The population involved in this study is not clear: did the authors compare STemi patients with patients that underwent staged procedures?

Why did the authors report "data not shown"?

The number of patients is not enough to consider the strategy mentioned safety.

In literature there a other trials comparing LVEDP vs UFR guided hydration with a lot of patients or comparing the type of contrast used, what is your novelty?

Minor issues: need to check punctuation.

Reviewer #2: In this small sample retrospective study, the authors investigated the effectiveness of a comprehensive kidney protection (KPS) strategy compared to usual care (UC) in reducing contrast-associated acute kidney injury (CA-AKI) in patients with an eGFR rate between 15-45 mL/min who received coronary angiography (CAG). The KPS approach resulted in lower contrast media exposure and a higher proportion of patients receiving a hydration schedule with saline administration guided by left ventricular end-diastolic pressure (LVEDP). Although there is conflicting data on using N-acetylcysteine (NAC) in preventing CA-AKI, some studies suggest that administering NAC with crystalloids before kidney insult onset could prevent CA-AKI. In this study, a significantly higher proportion (89.5%) of KPS patients received NAC than UC patients (21.6%). The authors should explain why the higher NAC administration in KPS patients did not correlate with a lower incidence of CA-AKI.

Reviewer #3: In the presented study, Jong CB and co-authors investigated the effect of a novel comprehensive kidney protection strategy (KPS) on the risk of developing contrast-associated acute kidney injury (CA-AKI). This was a single-center historical cohort study of 70 patients with moderate to advanced chronic kidney disease (eGFR 15-45 ml/min/1.73 m2) who had undergone invasive cardiac catheterization – coronary angiography (CAG): 19 patients received the KPS strategy and 51 received standard care. In the adjusted analysis, the KPS group had a significantly lower risk of CA-AKI than the standard treatment group (4% vs. 20.4%; odds ratio 0.19). This effect was consistent across different patients subgroups, including elderly patients, acute myocardial infarction, advanced CKD, high-risk category, left ventricular systolic dysfunction, and multivessel disease.

The manuscript is generally well written. Since the effect of iodinated contrast agents on the development of CA-AKI is still controversial, this study is of potential interest. However, I would have further comments and suggestions as the presented study has several limitations and problems that should be addressed.

Main Comments

1. Concept of contrast-associated acute kidney injury and prevention strategies

The concept and scientific evidence for CA-AKI has changed in recent years, as the causal association of contrast agents with AKI is not clear. Accordingly, the consensus-based recommendations for their prevention and treatment have evolved. This should be briefly outlined in the introduction section of the manuscript.

2. Study population

a. It is not clear from the data presented how patients were selected for the new KPS strategy compared to standard-of-care strategy for the prevention of CA-AKI. Based on which baseline characteristics? This could lead to selection bias, which is a significant limitation of the presented study.

b. The authors have presented a patient inclusion flowchart in Supplementary Figure 1. It appears that 17 patients were excluded from the standard treatment group because they received a KPS strategy during the CAG procedure. Why were these patients included in the standard treatment group in the first place and not in the KPS strategy group? This should be clarified.

3. Standard-of-care strategy

The authors should present the standard- of-care strategy for the prevention of CA-AKI in more detail. How were patients selected for standard treatment?

4. Pre-existing kidney disease vs. acute kidney injury

The authors should explain in more detail whether the patients with an eGFR of <45 mL/min/1.73 m2 were patients with moderate to advanced CKD (stages G3b and G4) or whether patients with AKI and an eGFR of <45 mL/min/1.73 m2 were also included. This should be clarified.

5. KPS protocol and contrast volume

According to the KPS protection protocol, a maximum contrast volume equal to or less than the last numerical eGFR value was used to minimize contrast administration during CAG. Did all patients undergo diagnostic CAG only or were there patients (in the KPS group and in the standard treatment group) who underwent coronary intervention with baloon angioplasty and stenting immediately following the diagnostic procedure? In this case, the contrast volume is much higher. This should be explained in more detail.

6. CA-AKI

As already mentioned, the causal attribution of contrast agents leading to AKI is not clear. Based on the follow-up of serum creatinine on days 1 and 3 after contrast exposure, the authors should also report and compare the proportions of patients with stable kidney function and a decrease in serum creatinine concentration of at least 0.3 mg/dL in both groups (and not just an increase in serum creatinine) during the first 3 days after contrast exposure (these data should be presented in Table 2).

7. Baseline patient characteristics and risk for CA-AKI

Based on the data presented in Table 1, a significantly higher proportion of patients in the standard-of-care group had multivessel disease and more complex coronary artery disease anatomy. Nearly 30% of patients in the standard group underwent immediate PCI compared to 10% of patients in the KPS group. In addition, patients in the standard group were less likely to receive isotonic saline protection, which is the standard-of-care for prevention of CA-AKI. All in all, it is not surprising that patients in the standard treatment group received a higher volume of contrast medium, had a higher baseline risk and thus a higher incidence of CA-AKI (independent of statistical adjustments). This is confirmed by the unadjusted baseline risk of CA-AKI, where patients in the standard treatment group had a numerically higher baseline risk of CA-AKI. Due to the small number of patients in both groups, all calculations of statistical differences and statistical adjustments could be associated with a type 2 statistical error.

8. Predicted and actual risks for CA-AKI

The authors should also compare the predicted and actual risks for CA-AKI in both patient groups (Supplementary Figure 2 shows only the data for the entire cohort).

9. The CA-AKI risk calculation

a. The authors should compare the risk of CA-AKI between groups in the unadjusted cohort (and not just in the IPTW-adjusted cohort).

b. Hydration with isotonic saline is one of the most important factors in preventing CA-AKI. This is also confirmed by the univariate analysis presented in Supplementary Table 1. Since less than 50% of patients in the standard treatment group were hydrated (compared to almost 90% in the KPS group), hydration status with isotonic saline should be included in all logistic regression analyzes (univariate and multivariate). A lower incidence of saline hydration might be associated with a higher incidence of CA-AKI in standard care patients.

c. Furthermore, if <50% of patients with moderate to advanced CKD receive hydration with isotonic saline, this cannot be considered the standard-of-care for the prevention of CA-AKI in high-risk patients.

10. Patient outcomes

The authors should present and compare the final outcomes in both groups of patients, not just short-term data on the incidence and risk of CA-AKI based on serum creatinine in the first 3 days after cardiac catheterization. I would be interested in data on differences in morbidity and mortality in both groups of patients, including data on the evolution of kidney function at last follow-up, progression of CKD and patient survival.

Other comments

1. Authors should refrain from using non-standardized definitions or terms for kidney disease/injury. The term kidney insufficiency (KI) should be replaced by chronic kidney disease (CKD) or acute kidney injury (AKI).

6. PLOS authors have the option to publish the peer review history of their article (what does this mean?). If published, this will include your full peer review and any attached files.

Reviewer #1: No

Reviewer #2: **Yes: **JESUS ARELLANO MARTINEZ

Reviewer #3: No

---

## [Author Response · Author response to Decision Letter 0]

27 Aug 2024

Response to Reviewers

Reviewer #1: Dear editor,

first of all thank you for asking me to review this manuscript about kidney protection strategy to lower contrast induced kidney injury. To date, this is a extremely hot topic.

For the regards the content:

1. Can the authors be more specific about patients enrolled? Are all ST elevation myocardial infarction patients? or not? The population involved in this study is not clear: did the authors compare STEMI patients with patients that underwent staged procedures?

Reply：

Thank you for this suggestion. In this study, we enrolled patients diagnosed with acute coronary syndrome, chronic coronary syndrome, and heart failure. Of the total participants, 9 patients presented with ST-elevation myocardial infarction (STEMI), and 7 patients underwent primary percutaneous coronary intervention (PCI), with 1 patient in the KPS group and 6 patients in the UC group. Additionally, one patient with STEMI underwent staged PCI in the KPS group, while another patient in the UC group underwent staged coronary artery bypass grafting (CABG).

For patients with STEMI, the reason for delayed revascularization was good distal perfusion of the culprit vessel, which was perfused by collateral circulation. It is important to note that the number of patients with STEMI was relatively small, with only 2 patients in the KPS group and 7 patients in the UC group, making it challenging to compare outcomes with those of patients who underwent staged procedures. We have revised the inclusion criteria in the Methods section, and a comparison of STEMI presentations is presented in the revised Table 1.

[Page 6, Line 82] The inclusion criteria were Visipaque 320 administration, eGFR of <45 mL/min/1.73 m2 during CAG, and CAG indication with acute coronary syndrome, chronic coronary syndrome, and heart failure.

Revised Table 1 Baseline demographics and clinical characteristics of the patients

Variable KPS

(n = 19) UC

(n = 51) P value

Clinical presentation 0.900

STEMI 2 (10.5) 7 (13.7) 

NSTEMI 6 (31.6) 19 (37.3) 

Heart failure 10 (52.6) 21 (41.2) 

US/CCS 1 (5.3) 4 (7.8) 

Why did the authors report "data not shown"?

Reply：

Thanks for your query. The “data not shown” mean the relevant data were not shown in the presented Table or Figure. To avoid misunderstandings, these sentences have been rephrased as:

[Page 16, line 190] Overall, the median Mehran 2 risk score was 8 (interquartile range [IQR]: 6–11.25), and approximately 51% of the patients were classified into high- and very high-risk categories.

[Page 20, line 265] Currently, all patients in the KPS group received ULCAG, and one-third of patients underwent staging procedures, such as zero contrast PCI and cardiac surgery, a few days later.

The number of patients is not enough to consider the strategy mentioned safety. 

Reply：

Thank you for your comments. We agree that the safety of this strategy is uncertain. The sentence has been revised as follows.

[Page 19, line 247] This modified protocol was feasible for emergency procedures and in patients with medically controlled severe valvular heart disease.

In literature there are other trials comparing LVEDP vs UFR guided hydration with a lot of patients or comparing the type of contrast used, what is your novelty?

Reply：

Thanks for your query. Several hydration strategies have been proposed, and their efficacy in preventing CA-AKI has been demonstrated in recent clinical trials. However, the contrast volume was higher in these clinical trials than that in our study. (Please refer to the table below)

Trials Age Estimated GFR Contrast volume History of heart failure

POSEIDON 71±9 48±9 104(84-187) 31 (16%)

Geng Qian et al 64±10 36 (23-48) 161±67 132 (100%)

HYDRA 68±10 73±27 135±94 36 (5.1%)

AVERT 71±9.3 45.6±10.7 85.6±50.5 89 (31.1)

REMEDIAL III 74±8 36±3 72±49 NA

STRENTGH 79.1±8.8 32±9 116±68 43 (35.2%)

NTUH-HC (KPS group) 73.4±10.6 29.1±8.1 7.5(5-11) 16 (84.2%)

An ultralow contrast strategy was recommended in the KPS group in our study (Table 2). Both estimated GFR and contrast volume were the lowest in our study, in comparison with other clinical trials. (Please refer to table above). To our knowledge, this is the first study to show the cumulative effect of LVEDP-guided hydration and ULCAG in reducing CA-AKI in patients with concomitant high-risk conditions and KI.

Minor issues: need to check punctuation.

Reply：

Thank you for this suggestion. The punctuation was checked and corrected.

Reviewer #2: In this small sample retrospective study, the authors investigated the effectiveness of a comprehensive kidney protection (KPS) strategy compared to usual care (UC) in reducing contrast-associated acute kidney injury (CA-AKI) in patients with an eGFR rate between 15-45 mL/min who received coronary angiography (CAG). The KPS approach resulted in lower contrast media exposure and a higher proportion of patients receiving a hydration schedule with saline administration guided by left ventricular end-diastolic pressure (LVEDP). Although there is conflicting data on using N-acetylcysteine (NAC) in preventing CA-AKI, some studies suggest that administering NAC with crystalloids before kidney insult onset could prevent CA-AKI. In this study, a significantly higher proportion (89.5%) of KPS patients received NAC than UC patients (21.6%). The authors should explain why the higher NAC administration in KPS patients did not correlate with a lower incidence of CA-AKI.

Reply：

Thank you for your comments. We would like to clarify a few points regarding our study of the association between NAC administration (acetylcysteine) and the risk of CA-AKI. 

First, we conducted an analysis using the entire cohort, including both the KPS and UC groups, to assess this association. The results suggested a protective effect of NAC administration with an odds ratio of 0.56, although this finding was not statistically significant. We attribute this lack of significance to the limited sample size in our study (S1 Table).

Second, in our primary analysis, which was performed on the IPTW-adjusted cohort, we found a significant reduction in the risk of CA-AKI in patients receiving KPS compared with those in the UC group. The odds ratio was 0.19 with a p-value of 0.009 (Figure 2). We believe that this difference was likely due to the higher frequency of NAC administration in the KPS group.

We appreciate your input and plan to address the limitations of our study, such as its small sample size.

Reviewer #3: In the presented study, Jong CB and co-authors investigated the effect of a novel comprehensive kidney protection strategy (KPS) on the risk of developing contrast-associated acute kidney injury (CA-AKI). This was a single-center historical cohort study of 70 patients with moderate to advanced chronic kidney disease (eGFR 15-45 ml/min/1.73 m2) who had undergone invasive cardiac catheterization – coronary angiography (CAG): 19 patients received the KPS strategy and 51 received standard care. In the adjusted analysis, the KPS group had a significantly lower risk of CA-AKI than the standard treatment group (4% vs. 20.4%; odds ratio 0.19). This effect was consistent across different patients subgroups, including elderly patients, acute myocardial infarction, advanced CKD, high-risk category, left ventricular systolic dysfunction, and multivessel disease.

The manuscript is generally well written. Since the effect of iodinated contrast agents on the development of CA-AKI is still controversial, this study is of potential interest. However, I would have further comments and suggestions as the presented study has several limitations and problems that should be addressed.

Main Comments

1. Concept of contrast-associated acute kidney injury and prevention strategies

The concept and scientific evidence for CA-AKI has changed in recent years, as the causal association of contrast agents with AKI is not clear. Accordingly, the consensus-based recommendations for their prevention and treatment have evolved. This should be briefly outlined in the introduction section of the manuscript.

Reply：

Thank you for your constructive suggestion. The Introduction section has been revised as follows:

[Page 4, line 56] The causal association of contrast agents with AKI remained controversial in recent years. The role of contrast agents in the development of AKI may be overestimated, especially in cases of multi-comorbidity risks. Therefore, this study stratified the risk of AKI according to a novel predictive model that included eight clinically important variables, with the exception of the contrast volume. Herein, a comprehensive kidney protection strategy (KPS) that combines the concepts of LVEDP-guided isotonic saline hydration, ULCAG, and a staged coronary revascularization procedure under suitable conditions was introduced.

2. Study population

a. It is not clear from the data presented how patients were selected for the new KPS strategy compared to standard-of-care strategy for the prevention of CA-AKI. Based on which baseline characteristics? This could lead to selection bias, which is a significant limitation of the presented study.

Reply：

Thank you for your comments. All patients referred to our study team for coronary angiography (CAG) underwent kidney protective strategies (KPS) if they had comorbidities and an estimated glomerular filtration rate (eGFR) of less than 45 mL/min/1.73 m2. Patients with kidney insufficiency (KI) who were referred to other interventionists and underwent similar procedures but did not receive KPS during CAG were categorized as the usual care (UC) group. The unintentional referral of patients may have contributed to the similar baseline characteristics between the two groups, except for the diagnosis of multivessel coronary artery disease, which was determined after invasive CAG. However, this may have introduced selection bias, and therefore, the Methods and Limitations sections have been revised to address this concern.

[Page 5, Line 68] All patients referred to our study team for CAG underwent KPS, while comorbidity with an eGFR of <45 mL/min/1.73 m2. In contrast, patients with KI who were referred to other interventionists and underwent similar procedures but did not receive KPS during CAG were categorized into the usual care (UC) group. This study enrolled patients from two databases: those registered in the catheterization laboratory database and those who underwent KPS during CAG performed by the study team. The enrollment periods of the two databases overlapped, and 17 patients in the catheterization laboratory database were reclassified into the KPS group because they had undergone KPS during CAG. (S1 Fig.)

[Page 20, Line 272] This study has few limitations. First, the influence of residual confounding factors and selection and indication biases are unavoidable in observational studies.

b. The authors have presented a patient inclusion flowchart in Supplementary Figure 1. It appears that 17 patients were excluded from the standard treatment group because they received a KPS strategy during the CAG procedure. Why were these patients included in the standard treatment group in the first place and not in the KPS strategy group? This should be clarified.

Reply：

Thank you for this suggestion. The Methods section has been revised as follows:

[Page 5, Line 68] All patients referred to our study team for CAG would undergo KPS, while comorbidity with an eGFR of <45 mL/min/1.73 m2. In contrast, patients with KI who were referred to other interventionists and underwent similar procedures but did not receive KPS during CAG were categorized into the usual care (UC) group. This study enrolled patients from two databases: those registered in the catheterization laboratory database and those who underwent KPS during CAG performed by the study team. The enrollment periods of the two databases overlapped, and 17 patients in the catheterization laboratory database were reclassified into the KPS group because they had undergone KPS during CAG. (S1 Fig.)

3. Standard-of-care strategy

The authors should present the standard- of-care strategy for the prevention of CA-AKI in more detail. How were patients selected for standard treatment?

Reply：

Thank you for this suggestion. Patients with KI who were referred to an interventionist other than our study team but did not receive KPS during CAG were categorized into the usual care (UC) group. The Methods section has been revised, and the UC strategy has been introduced in more detail. Notably, the strategy for CA-AKI prevention in the UC group was not standardized in all patients, which reflects the real-world situation in the daily catheterization room. For example, the rate of hydration was significantly lower in the UC group, wherein two-thirds of patients had a history of HF. Most non-hydrated patients may be concerned about fluid overload, which reflects the real-world situation in clinical practice. Therefore, we used the term UC instead of the standard-of-care in our study.

[Page 8, Line 115]

UC strategy

In contrast, patients receiving UC were hydrated with an isotonic saline infusion at a constant rate of 0.5 or 1 mL/kg/h in those with or without heart failure (HF) before and after 12 h of contrast exposure [8]. Notably, the strategy for CA-AKI prevention in the UC group was not standardized in all patients, which reflects the real-world situation in the daily catheterization room. Some patients may not have received saline hydration because of concerns regarding fluid overload. Minimum contrast volume administration and decision on the timing of coronary revascularization were not standardized; they depended on the operator’s discretion.

4. Pre-existing kidney disease vs. acute kidney injury

The authors should explain in more detail whether the patients with an eGFR of <45 mL/min/1.73 m2 were patients with moderate to advanced CKD (stages G3b and G4) or whether patients with AKI and an eGFR of <45 mL/min/1.73 m2 were also included. This should be clarified.

Reply：

Thank you for your insightful comment. The inclusion criteria of our study were Visipaque 320 administration and eGFR <45 mL/min/1.73 m2 during CAG. We have rephrased the term of CKD to ‘eGFR at cath’ as half of the patient have not data of baseline eGFR (determined by prior 3-6 months of CAG). The baseline eGFR data are provided in revised Table 1. Owing to the lack of baseline eGFR data, the proportion of patients with AKI or CKD was uncountable. However, we have presented the follow-up kidney function status (i.e., improved, stable, and deteriorated) in the revised Table 2, and nearly one-third of the patients had improved kidney function in both groups without significant differences. Based on this result, we assumed that the proportion of AKI in both groups might be similar and have less impact on the risk of CA-AKI. Consistently, the kidney function reported in Mehran’s study was the serum creatinine measured at admission or on the morning of the PCI procedure; the creatinine level prior to 3 months of the procedure was not reported. Therefore, the kidney function in the current study was consistent with the Mehran 2 risk scoring system. However, this important issue has been mentioned in the Limitations section. Furthermore, there was only one patient in the KPS group with a baseline eGFR <30 ml/min/1.73 m², rendering the subgroup analysis inapplicable.

Revised Table 1 Baseline demographics and clinical characteristics of the patients

Variable KPS

(n = 19) UC

(n = 51) P value

Serum creatine at cath, mg/dL 2.4 ± 0.7 2.3 ± 0.7 0.696

Mean eGFR at cath, ml/min/1.73 m2 29.1 ± 8.1 30.0 ± 8.7 0.707

eGFR 30-45 at cath, mL/min/1.73 m2 10 (52.6) 27 (52.9) 1.000

eGFR <30 at cath, mL/min/1.73 m2 9 (47.4) 24 (47.1) 

Baseline eGFR, mL/min/1.73 m2 (n = 35) 36.9 ± 11.7 (n = 5) 36.9 ± 11.7 (n = 30) 0.680

Revised Table 2 Characteristics of KPS and CA-AKI in the patients

Variable KPS

(n = 19) UC

(n = 51) P value

Kidney function after day 1-3 of contrast exposure 0.380

Improved 6 (31.6) 16 (31.4) 

Stable 12 (63.2) 25 (49.0) 

AKI (deteriorated) 1 (5.3) 10 (19.6) 

[Page 21, Line 277] Third, half of the patients did not have baseline eGFR data, which implies that some of th

---

## [Decision Letter · Decision Letter 1]

4 Oct 2024

PONE-D-24-15647R1Kidney protection strategy lowers the risk of contrast-associated acute kidney injuryPLOS ONE

Dear Dr. Jong,

Thank you for submitting your manuscript to PLOS ONE. After careful consideration, we feel that it has merit but does not fully meet PLOS ONE’s publication criteria as it currently stands. Therefore, we invite you to submit a revised version of the manuscript that addresses the points raised during the review process.

We look forward to receiving your revised manuscript.

Kind regards,

Eyüp Serhat Çalık

Academic Editor

PLOS ONE

Journal Requirements:

Additional Editor Comments:

We thank the authors for making the suggested revisions and necessary changes to the manuscripts. Your manuscript has been reviewed again, a few minor suggestions are available below. Good luck

Reviewers' comments:

Reviewer's Responses to Questions

**Comments to the Author**

1. If the authors have adequately addressed your comments raised in a previous round of review and you feel that this manuscript is now acceptable for publication, you may indicate that here to bypass the “Comments to the Author” section, enter your conflict of interest statement in the “Confidential to Editor” section, and submit your "Accept" recommendation.

Reviewer #1: All comments have been addressed

Reviewer #3: All comments have been addressed

2. Is the manuscript technically sound, and do the data support the conclusions?

Reviewer #1: Partly

Reviewer #3: Yes

3. Has the statistical analysis been performed appropriately and rigorously? 

Reviewer #1: Yes

Reviewer #3: Yes

4. Have the authors made all data underlying the findings in their manuscript fully available?

Reviewer #1: Yes

Reviewer #3: Yes

5. Is the manuscript presented in an intelligible fashion and written in standard English?

Reviewer #1: Yes

Reviewer #3: Yes

6. Review Comments to the Author

Reviewer #1: I still have two minor issues:

1- please refer to the only iso-osmolar contrast agent as iodixanol instead of its commercial name;

2- it is not clear if the highest sCr value along the first 72 hours has been considered to adjudicate CI-AKI, or if sCr has been evaluated on days 1 and 3 (line 125 page 8); in this last case it should be considered a further limitation.

Reviewer #3: Thanks for the extensive revision and clarifications. In my opinion, all major concerns were addressed.

7. PLOS authors have the option to publish the peer review history of their article (what does this mean?). If published, this will include your full peer review and any attached files.

Reviewer #1: **Yes: **Fortunato Iacovelli

Reviewer #3: No

---

## [Author Response · Author response to Decision Letter 1]

7 Oct 2024

Reviewer #1: 

I still have two minor issues:

1- please refer to the only iso-osmolar contrast agent as iodixanol instead of its commercial name;

Reply:

Thank you for your suggestion. The Methods section has been rephrased as:

[Page 5, line 76–78] "At our institute, iodixanol, a nonionic, iso-osmolar contrast medium, was administered to patients with KI during CAG. The inclusion criteria were iodixanol administration, an eGFR of <45 mL/min/1.73 m2 during CAG,"

2- it is not clear if the highest sCr value along the first 72 hours has been considered to adjudicate CI-AKI, or if sCr has been evaluated on days 1 and 3 (line 125 page 8); in this last case it should be considered a further limitation.

Reply:

Thank you for your comment. All serum creatinine follow-ups on days 1 and 3 were conducted within the monitoring period of the first 72 hours after contrast exposure. The highest post-CAG creatinine level was recorded to determine the CA-AKI incidence. However, fluctuations in serum creatinine levels during this period may occur, and some data might go undetected, particularly in situations without frequent monitoring. This limitation could lead to an underestimation of the incidence of CA-AKI. The limitation section has been revised as follows:

[Page 20, line 277–280] "Fourth, fluctuations in post-CAG serum creatinine levels within the 72-hour monitoring period may occur, and some data might go undetected, particularly in situations without frequent monitoring. This may lead to an underestimation of the incidence of CA-AKI.".

---

## [Editor Report · Decision Letter 2]

10 Oct 2024

Kidney protection strategy lowers the risk of contrast-associated acute kidney injury

PONE-D-24-15647R2

Dear Dr. Jong,

We’re pleased to inform you that your manuscript has been judged scientifically suitable for publication and will be formally accepted for publication once it meets all outstanding technical requirements.

Kind regards,

Eyüp Serhat Çalık

Academic Editor

PLOS ONE

Additional Editor Comments (optional):

I would like to thank the authors for their appropriate responses to the suggestions and revisions to their manuscripts. I wish them success.
---

## [Editor Report · Acceptance letter]

15 Oct 2024

PONE-D-24-15647R2 

PLOS ONE

Dear Dr. Jong, 

I'm pleased to inform you that your manuscript has been deemed suitable for publication in PLOS ONE. Congratulations! Your manuscript is now being handed over to our production team.

Kind regards, 

on behalf of

Dr. Eyüp Serhat Çalık 

Academic Editor

PLOS ONE